# Utility of Fatty Liver Index to predict reversion to normoglycemia in people with prediabetes

**Carla Busquets-Cortés**[1,2,3], **Miquel Bennasar-Veny**[1,4,5]*, **Ángel Arturo López-González**[2,6], **Sergio Fresneda**[1,4,5], **Manuela Abbate**[3], **Aina M. Yáñez**[3,4,5]

**1** Research Group on Evidence, Lifestyles & Health, Research Institute on Health Sciences (IUNICS), University of the Balearic Islands, Palma, Spain, **2** Escuela Universitaria ADEMA, Palma, Illes Balears, Spain, **3** Global Health and Human Development Research Group, University of the Balearic Islands, Palma, Spain, **4** Instituto de Investigación Sanitaria Illes Balears (IdISBa), Palma, Illes Balears, Spain, **5** Department of Nursing and Physiotherapy, Balearic Islands University, Palma, Illes Balears, Spain, **6** Prevention of Occupational Risks in Health Services, Balearic Islands Health Service, Palma, Spain

* miquel.bennasar@uib.es

**Data Availability Statement:** All relevant data are within the manuscript.

**Funding:** The authors received no specific funding for this work.

## Abstract

### Background

Fatty Liver Index (FLI) is strongly associated with changes in glycemic status and incident Type 2 Diabetes (T2D). The probability of reverting to normoglycemia from a state prediabetes could be determined by FLI, however such relationship remains poorly understood.

### Aim

To determine the clinical interest of using FLI to estimate prediabetes reversion at 5 years in patients with impaired fasting plasma glucose at baseline, and identify those factors associated with changes in FLI, that could contribute to the reversion of prediabetes.

### Methods

This 5-year cohort study included 16,648 Spanish working adults with prediabetes. Prediabetes was defined as fasting plasma glucose (FPG) between 100 and 125 mg/dl according to the ADA criteria, while prediabetes reversion was defined as a FPG <100 mg/dL. The population was classified as: FLI <30 (no hepatic steatosis), FLI 30–59 (intermediate status), and FLI ≥60 (hepatic steatosis).

### Results

At 5 years follow-up, 33.7% of subjects reverted to normoglycemia (annual rate of 6.7%). The adjusted binomial logistic regression model showed that scoring FLI <30 (OR 1.544; 95% CI 1.355–1.759), performing at least 150 min/week of physical activity (OR 4.600; 95% CI 4.088–5.177) and consuming fruits and vegetables daily (OR 1.682; 95% CI 1.526–1.855) were associated with the probability of reverting form prediabetes to normoglycemia. The ROC curve for prediction of reversion showed that FLI (AUC 0.774;95% CI 0.767–0.781) was a better predictor than FPG (AUC 0.656; 95% CI 0.648–0.664).

**Competing interests:** The authors have declared that no competing interests exist.

## Conclusions

Regular physical activity, healthy dietary habits and absence of hepatic steatosis are independently associated with the probability of reversion to normoglycemia in adult workers with prediabetes at baseline. Low FLI values (especially FLI< 30) may be useful to predict the probability of prediabetes reversion, especially in active subjects with healthy eating habits, and thus identify those who might benefit from early lifestyle intervention.

## Introduction

Diabetes is a major public health problem that is approaching epidemic proportions worldwide [1]. In the natural history of the disease most patients experience a phase of prediabetes characterized by fasting plasma glucose (FPG) concentrations higher than normal but below the threshold for the diagnosis of type 2 diabetes (T2D) [2]. According to the International Diabetes Federation (IDF), 463 million individuals aged 20–79 years (6.0% of the worldwide population) are estimated to have prediabetes [3]. The annual rates of progression to T2D for this population ranges between 5–10% [4], and the lifetime risk is over 70%, especially in overweight and obese individuals [5]. Nevertheless, depending on age, sex, ethnicity, geographical area, social class, and criteria used to define prediabetes [6], between 20% and 50% of people with prediabetes do not develop T2D and revert back to normoglycemia. Furthermore, while prediabetes increases the risk of cardiovascular disease and mortality [7], its reversion is related to an amelioration of cardiovascular risk factors [8].

Non-alcoholic fatty liver disease (NAFLD) is currently the most common cause of liver disease worldwide with a global prevalence of around 25%, which may vary among countries and ethnicities [9]. NAFLD is characterized by free fatty acids and triglycerides infiltration in the hepatocytes, not related to significant alcohol intake [10]; its development is associated with poor lifestyle choices such as poor eating habits and a sedentary behavior [11], as well as visceral obesity, insulin resistance and the metabolic syndrome [12,13]. Increasing epidemiological evidence suggests that NAFLD is a strong independent risk factor for the development of T2D [14,15]. Regardless of stages of hepatic steatosis [16] incident T2D is increased more than 2-fold in subjects with NAFLD compared to those without NAFLD [17], suggesting that the presence or absence of NAFLD, especially in a state of prediabetes, could strongly determine the risk for T2D.

The gold standard for diagnosing NAFLD and determine the stage of the disease, is by liver biopsy [18], however performing such procedure on every patient with suspected NAFLD is unviable for practical, safety and economic reasons. To diagnose NAFLD in its early stages and with reduced health costs, clinical and laboratory-based biomarkers have been used in predictive models [12,13,16,17,19–21]. Concretely, the Fatty Liver Index (FLI), a validated risk score system based on routine measurements of triglycerides (TG) and gamma-glutamyl transferase (GGT) concentrations, waist-circumference (WC) and body mass index (BMI), accurately identifies NAFLD and hepatic steatosis in the general population [20,22]. Moreover, like NAFLD, FLI correlates with insulin resistance and metabolic syndrome [21], and it significantly predicts the risk of T2D in people with prediabetes. The PREDAPS study, conducted in 1,142 Spanish adults with prediabetes attending primary care centers, demonstrated the utility of FLI to predict T2D development after 5 years of follow-up [23]. A recent study pointed out that low baseline FLI scores are independently associated with prediabetes reversion [13], nonetheless, the rate of reversion to normoglycemia from prediabetes determined by FLI in

patients with impaired glucose metabolism remains poorly understood. According to epidemiological and clinical evidence, the utilization of FLI during routine assessments in people with prediabetes contributes to the early targeting, and therefore treating, those at increased risk of developing T2D. It is likely that FLI could also predict reversion to normoglycemia in prediabetes, however more evidence is needed.

We aimed to determine the clinical interest of using FLI to estimate reversion to normoglycemia in a large cohort of service workers with baseline prediabetes during a 5-year follow-up. Factors associated to changes in FLI were also taken into consideration.

## Materials and methods

The present cohort study included 16,648 male and female Spanish adults, aged between 20 and 65 years, presenting prediabetes at baseline, and who worked in the service sector. The study methods have been described in detail previously [24]. Briefly, participants were carefully selected from a population of 234,995 potentially suitable individuals who underwent periodic occupational health assessments between 2012 and 2013. Inclusion criteria were age between 20 and 65 years, and FPG between 100 and 125 mg/dL. Exclusion criteria were history of previously diagnosed diabetes, current treatment with oral antidiabetic or systemic glucocorticoid, FPG ≥126 mg/dL or HbA1c ≥6.5% at baseline, cancer treatment in the preceding 5 years, anemia (hematocrit <36% in men and <33% in women), and pregnancy. At baseline, all subjects underwent standard health examinations, anthropometric measurements, and metabolic tests. Follow-up examinations were performed after a period of 5 years, in 2017 and 2018.

### Data collection and definition of variables

Sociodemographic and lifestyle characteristics were collected at baseline by means of a questionnaire. Specifically, participants were asked to report if they performed at least 150 min/ week of moderate and/or vigorous exercise (according to World Health Organization (WHO) recommendations) and if consumed vegetables and fruits daily. Smoking habits were also assessed, and individuals were categorized as "smoker", "former smoker", or "never smoker". Social class was defined according to the Spanish Epidemiology Society classification [25]. In general terms, Class I (upper class) includes executives, managers, and qualified professionals; Class II (middle class) includes intermediate occupations and employees; and Class III (lower class) includes manual workers.

All anthropometric measurements were taken at baseline and at 5 years according to the guidelines and recommendations of the International Standards for Anthropometric Assessment (ISAK) manual [26], and performed by qualified technicians or trained researchers to minimize coefficients of variation. Body weight was measured to the nearest 0.1 kg using an electronic scale (Seca 700 scale, Hamburg); height was measured to the nearest 0.5 cm using a stadiometer (Seca 220 Telescopic Height Rod for Column Scales, Hamburg); and BMI was calculated as weight (kg) divided by height (m) squared ($kg/m^2$). Obesity was defined as BMI ≥ 30.0 $kg/m^2$, in agreement with WHO criteria. Blood pressure was measured in triplicate, with a one-minute gap between measurements, using an electric and calibrated sphygmomanometer (OMRON M3, Healthcare Europe, Spain), with the patient in a supine position after 10 minutes rest. The mean of the three measurements was recorded.

Blood samples were taken at baseline and 5 years. Venous blood samples were collected from the antecubital vein after a 12h overnight fast in suitable vacutainers without anticoagulant to obtain serum. Serum concentrations of glucose, TG, GGT and cholesterol were

measured by standard procedures using a Beckman Coulter SYNCHRON CX® 9 PRO clinical system (La Brea, CA, USA).

Prediabetes was defined as FPG between 100 and 125 mg/dl according to the ADA criteria. Incident T2D was defined as FPG $\geq$ 126 mg/dl or the initiation of anti-hyperglycemic treatment at follow-up. Prediabetes reversion was defined as a FPG value <100 mg/dL at follow-up.

### FLI as a surrogate measure of fatty liver

The FLI was calculated based on measurements of TG, GGT, BMI and WC using the following formula [15]:

$$Fatty\ Liver\ Index(FLI) = e^y/(1 + e^y) \times 100$$

*Where*

$$y = 0.953 \times ln(TG) + 0.139 \times BMI + 0.718 \times ln(GGT) + 0.053 \times WC - 15.745$$

TG indicates triglyceride concentration, measured as mg/dl; BMI indicates body mass index, measured as kg/m$^2$; GGT indicates γ-glutamyl transpeptidase, measured as U/l; and WC indicates waist circumference, measured in cm.

FLI values range from 0 to 100 and show good diagnostic accuracy in detecting fatty liver, with an Area Under the Curve (AUC) of 0.85 and a 95% confidence interval (CI) of 0.81–0.88 [10,15]. FLI <30 has been found to rule out steatosis with a sensitivity of 87% and a specificity of 64%, whereas FLI $\geq$60 is indicative of the presence of steatosis with a sensitivity of 61% and specificity of 86% [22]. FLI values between 30 and 59 indicate indeterminate risk, and fatty liver might neither be confirmed nor ruled out. FLI scores have been validated by comparison with the results of liver ultrasound and nuclear magnetic resonance spectroscopy [27]. Accordingly, participants were classified into three categories: FLI <30 (low risk for NAFLD), FLI 30–59 (indeterminate risk for NAFLD) and FLI $\geq$60 (high risk for NAFLD).

### Statistical analyses

Continuous variables are expressed as means (± SDs) and were compared by Student's t-test, whereas categorical variables are expressed as n (%) and were compared by chi-square ($\chi^2$) tests. Multivariate logistic regression analyses were performed to calculate odds ratios (ORs) and corresponding 95% confidence intervals (CI) of the reversion from prediabetes to normoglycemia, while adjusting for potential confounders that showed a significant association in univariate analysis. Finally, the receiver operating characteristic (ROC) curve analysis was used to determine the predictive ability of FLI as a continuous variable to classify subjects with prediabetes reverting to normoglycemia.

Analyses were performed using the Statistical Package for the Social Sciences (SPSS) software version 25.0 (IBM Company, New York, NY, USA) for Windows. Statistical tests were two-sided, and p values <0.05 were considered statistically significant.

### Ethical considerations

All the procedures in the study protocol were in accordance with the Declaration of Helsinki for research on human participants and were approved by the Balearic Ethical Committee of Clinical Research (Ref. No: CEI-IB-1887). All participants were properly informed of the aim and requirements of the study before giving their written consent to participate.

## Results

The current analysis included 12,080 (72,6%) males and 4,568 (27,4%) females subjects. Mean age ± SD was 44, 5 ± 9.8 years. Of the 16,648 subjects with prediabetes at baseline, 5,604 (33,7%; 3,907 males and 1,697 females) reverted to normal levels of blood glucose at 5 years follow-up. At follow-up 35,5% of the population exhibited FLI values above 60, while 38,5% had FLI <30.

Demographic, lifestyle, and clinical characteristics of the study sample as a whole and stratified by reversion to normoglycemia are shown in Table 1.

When considering the whole sample, most subjects belonged to social class III (78,9%), 42,8% were overweight, while 26,8% were obese. As for smoking habits, 45,9% never smoked, while 32,7% were current smokers. When dividing subjects according to reversion to normoglycemia, amongst those who reverted from a prediabetic state to normal values of FPG, 64,4% presented FLI values below 30, while only 9,9% had FLI ≥60. On the other hand, 48,2% of those who did not revert presented FLI values ≥60. Subjects who did not revert also presented

**Table 1. Anthropometric characteristics and biochemical parameters of subjects according to reversion to normal glycemia.**

| Characteristics | Total (n = 16,648) | No Reversion (n = 11,044) 66,33% | Reversion (n = 5,604) 33,66% | P value |
|---|---|---|---|---|
| FLI <30 | 6,421 (38.6%) | 2,799 (25.3%) | 3,622 (64.4%) | < 0.001 |
| FLI 30–59 | 4,318 (25.9%) | 2,921 (26.4%) | 1,397 (32.4%) | < 0.001 |
| FLI ≥60 | 5,909 (35.5%) | 5,324 (48.2%) | 585 (9.9%) | < 0.001 |
| Age (years) | 44.5 ± 9.8 | 45.9 ± 9.4 | 41.7 ± 10.3 | < 0.001 |
| Sex (male) | 12,080 (72.6%) | 8,173 (67.7%) | 3,907 (32.3%) | < 0.001 |
| Social class (ref. I)[a] | | | | 0.074 |
| I | 741 (4.5%) | 492 (4.5%) | 249 (4.4%) | |
| II | 2,779 (16.7%) | 1,853 (16.8%) | 926 (16.5%) | |
| III | 13,128 (78.9%) | 8,699 (78.8%) | 4,429 (79.9%) | |
| BMI (kg/m$^2$) | 27.7 ± 4.8 | 29.1 ± 4.9 | 24.9 ± 3.2 | < 0.001 |
| BMI categories | | | | < 0.001 |
| Normal weight | 5,046 (30.4%) | 2,050 (18.6%) | 2,996 (53.4%) | |
| Overweight | 7,115 (42.8%) | 4,830 (43.8%) | 2,285 (40.8%) | |
| Obese | 4,460 (26.8%) | 4,145 (37.6%) | 315 (5.6%) | |
| WC (cm) | 86.1 ± 9.45 | 88.9 ± 9.7 | 83.2 ± 9.2 | < 0.001 |
| Triglycerides (mg/dL) | 137.6 ± 106.4 | 154.0 ± 118.2 | 105.4 ± 67.2 | < 0.001 |
| Glucose (mg/dL) | 106.2 ± 5.8 | 107.2 ± 6.1 | 104.4 ± 4.5 | < 0.001 |
| Cholesterol (mg/dL) | 202.3 ± 38.1 | 207.9 ± 37.9 | 191.5 ± 36.0 | < 0.001 |
| GGT (UI/l) | 44.2 ±5 3.7 | 50.8 ± 57.1 | 31.2 ± 43.5 | < 0.001 |
| SBP (mmHg) | 127.9 ± 16.7 | 129.8 ± 16.8 | 124.0 ± 15.9 | < 0.001 |
| DBP (mmHg) | 78.3 ± 11.0 | 79.8 ± 10.9 | 75.4 ± 10.5 | 0.795 |
| PA (≥150 min/week)[a] | 6,892 (41.4%) | 2,549 (23.3%) | 4,343 (77.5%) | < 0.001 |
| Diet (daily fruits and vegetables)[a] | 6,771 (40.7%) | 2,919 (26.4%) | 3,852 (68.7%) | < 0.001 |
| Smoking habit[a] | | | | < 0.001 |
| Never | 7,645 (45.9%) | 4,947 (44.8%) | 2,698 (48.1%) | |
| Former | 3,549 (21.3%) | 2,641 (23.9%) | 908 (16.2%) | |
| Current | 5,454 (32.7%) | 3,456 (31.3%) | 1,998 (35.7%) | |

Results are reported as mean ± SD or n (%).

[a] Baseline data.

FLI, fatty liver index; BMI, body mass index; WC, waist circumference; GGT, γ-glutamyl transpeptidase; SBP, systolic blood pressure; DBP, diastolic blood pressure PA, physical activity.

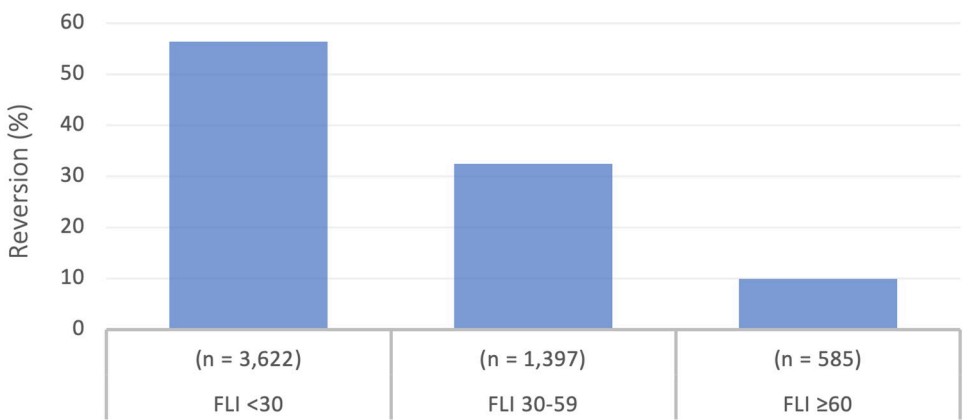

**Fig 1. Reversion from prediabetes to normoglycemia according to FLI categories.**

significantly higher BMI and WC, higher TG, FPG, cholesterol, and GGT, and higher levels of SBP than subjects who reverted to normoglycemia at 5 years follow-up. They were also less likely to performed at least 150 min per week of PA and consume fruits and vegetables every day at baseline.

Of the 16,648 subjects with prediabetes, 3,706 (22.2%) progressed to T2D at 5 years, which corresponded to an annual rate of 4.5%, while the annual rate of reversion to normoglycemia was of 6.7%. According to FLI categories, reversion to normoglycemia occurred in 3,622 (54.6%) subjects with FLI <30; in 1,397 (32.4%) subjects with FLI values <60; and in 585 (9.9%) subjects with FLI ≥60 (Fig 1).

Participants who reverted from prediabetes to normoglycemia had significantly lower scores of FLI than those who did not (Fig 2).

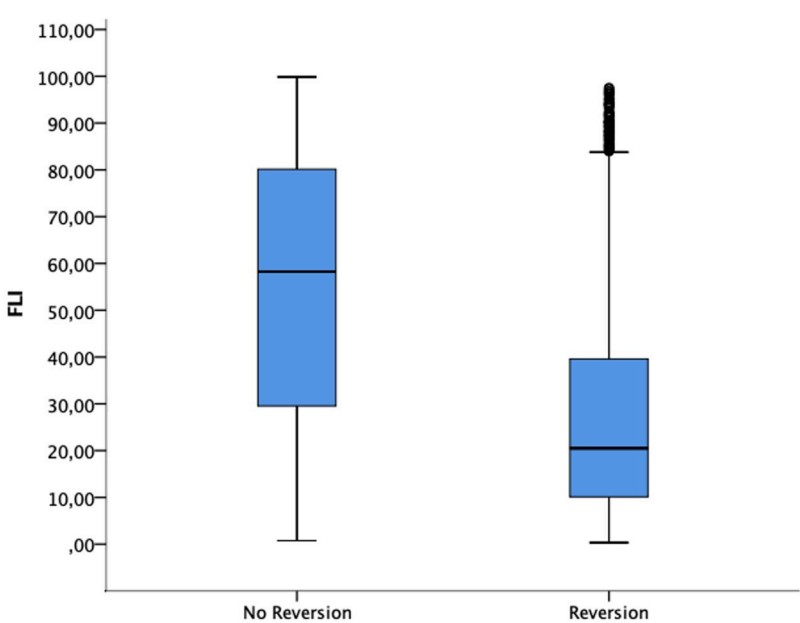

**Fig 2. Fatty Liver Index values in reversion and no reversion groups.**

**Table 2. Odds ratio for reversion from prediabetes to normoglycemia in patients with prediabetes.**

| Variables | OR crude (95% CI) | OR adjusted (95% CI) |
|---|---|---|
| Age | 0.962 (0.959–0.964) | 0.981 (0.977–0.985) |
| Men (Ref: women) | 0.827 (0.779–0.878) | 1.011 (0.917–1.114) |
| Social class (Ref: I) | | |
| II | 1.192 (1.056–1.346) | 1.047 (0.850–1.289) |
| III | 1.100 (0.935–1.075) | 1.200 (0.992–1.452) |
| PA (≥150 min/week) | 8.134 (8.134–9.190) | 4.600 (4.088–5.177) |
| Diet (daily fruits and vegetables) | 5.462 (5.153–5.790) | 1.682 (1.526–1.855) |
| Smoking habits (Ref: never smoker) | | |
| Former | 0.533 (0.488–0.582) | 0.797 (0.714–0.889) |
| Current | 0.916 (0.864–0.972) | 0.950 (0.868–1.041) |
| BMI | 0.766 (0.759–0.773) | 0.918 (0.903–0.932) |
| SBP | 0.978 (0.976–0.979) | 1.002 (0.999–1.005) |
| FPG | 0.921 (0.916–0.926) | 0.908 (0.900–0.915) |
| FLI (Ref: FLI ≥60) | | |
| FLI 30–59 | 4.353 (3.913–4.842) | 1.399 (1.188–1.647) |
| FLI <30 | 11.777 (10.671–12.997) | 1.544 (1.355–1.759) |

PA, physical activity; BMI, body mass index; SBP, systolic blood pressure; FPG, fasting plasma glucose; FLI, fatty liver index.

Table 2 shows bivariate associations of sociodemographic and clinical variables with reversion to normoglycemia in people with prediabetes. Univariate analysis showed that absence of hepatic steatosis (FLI <30) (OR 11.777; 95% 10.671–12.997) was strongly associated with reversion to normoglycemia compared to FLI ≥60. Moreover, PA ≥150 min/week (OR 8.134; 95% CI 8.134–9.190) and consuming fruits and vegetables daily (OR 5.462; 95% CI 5.153–5.790) were also significantly associated with the reversion. On the other hand, males compared to females (OR 0.827; 95% CI 0.779–0.878), subjects with increased age (OR 0.962; 95% 0.959–0.964), BMI (OR 0.766; 95% CI 0.759–0.773) and FPG (OR 0.921; 95% CI 0.916–0.926), and former (OR 0.533; 95% CI 0.488–0.582) and current (OR 0.916; 95% CI 0.864–0.972) smokers, had lower probability of prediabetes reversion.

The adjusted binomial logistic regression model showed virtually unchanged results. FLI <30 remained independently associated with reversion to normoglycemia (OR 1.544; 95% CI 1.355–1.759). Scoring a FLI between 30–59 (OR 1.399; 95% CI 1.188–1.647), performing at least 150 min/week of PA (OR 4.600; 95% CI 4.088–5.177) and consuming fruits and vegetables daily (OR 1.682; 95% CI 1.526–1.855) also remained significantly associated with the probability of reverting form prediabetes to normoglycemia, while no association was found between prediabetes reversion and age, male sex, current and former smoking, BMI, SBP and FPG.

As showed in Fig 3, FLI predicted reversion to normoglycemia better than FPG. The area under the curve (AUC) for the ROC analysis was 0.656 (95% CI 0.648–0.664) for FPG, and 0.774 (95% CI 0.767–0.781) for FLI.

## Discussion

In the present cohort study, about one third of subjects with FPG-defined prediabetes at baseline reverted to normoglycemia after 5 years of follow-up. Lower FLI categories were significantly associated with prediabetes reversion. In multivariate analysis FLI remained a

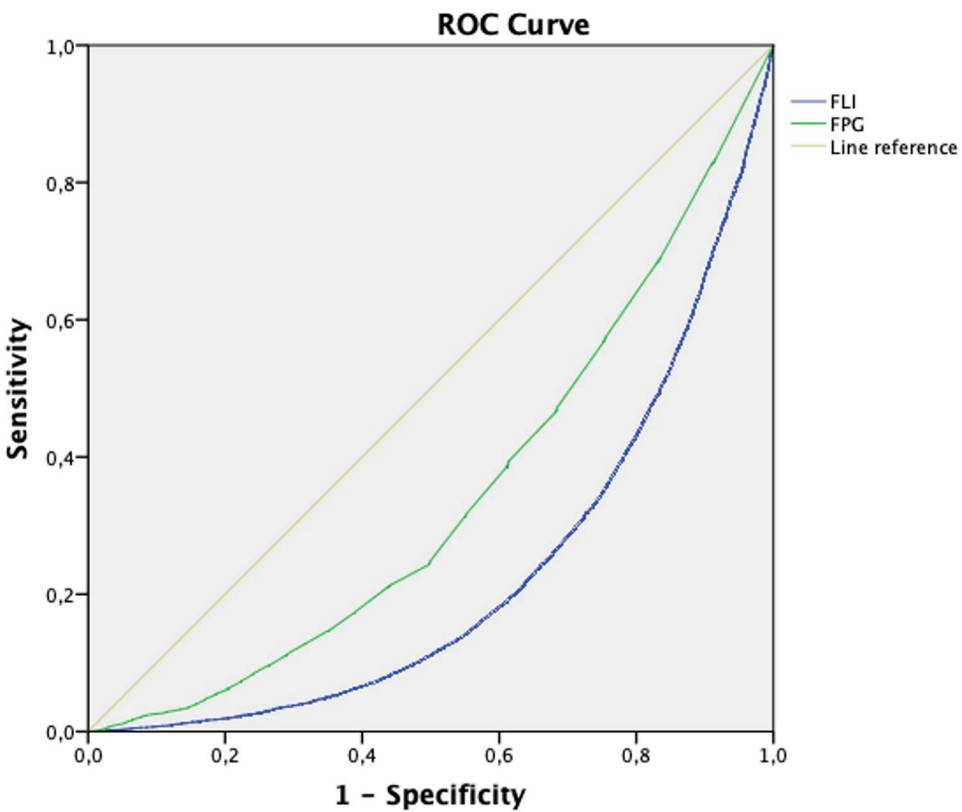

**Fig 3. ROC curve for prediction of reversion from prediabetes to normoglycemia by FLI and FPG.**

significant predictor of reversion to normoglycemia after accounting for potential confounders. Subjects who reverted to normoglycemia presented a better cardiometabolic profile, lower rates of NAFLD (assessed by FLI) and higher adherence to PA and dietary recommendations than subjects who did not revert. Moreover, performing PA ≥150 min/week and eating fruits and vegetables daily were also directly associated with an increased probability of reverting to normoglycemia within 5 years. Lastly, the AUC estimates of FLI predicted prediabetes reversion with better accuracy than FPG.

The reversion rate observed in our study is consistent with previously reported data. A Japanese population study which included patients 5 years older than our sample, and which used HbA1c 5.7–6.4% as a criterion to define prediabetes, reported a reversion rate of 32% over a 5 year period [28]. Other studies, using FPG as a screening criterion, showed reversion rates ranging from 28 to 45% over the same follow-up period [29]. Moreover, a recent Cochrane systematic review [30] including 47 studies on prediabetes reversion, showed regression rates between 33% and 59% at one to five years follow-up, and between 17% and 42% at 6 to 11 years follow-up.

Liver fat accumulation has been recognized as a risk factor for the development of T2D. In a longitudinal study [29] of 129 Swedish adults with biopsy proven NAFLD and elevated levels of serum transaminases, the prevalence of T2D and impaired glucose tolerance went from 8.5% at baseline, to 80% at the end of a 14 year period. Similarly, in a Korean study including 13,218 subjects without T2D at baseline and followed-up during 5 years, those who progressed to more severe stages of NAFLD had a significantly increased risk of developing T2D (OR 2.49 95% CI 1.49–4.14) compared to those that resolved NAFLD over the same period [31]. In the

Spanish PREDAPS study, a FLI >60 was independently associated with T2D incidence after 5 years of follow-up, independently of sex, age and educational level [23]. In the Diabetes Remission Clinical Trial (DiRECT), early onset T2D was associated with increased liver fat accumulation [32]. On the other hand, diabetes remission with concomitant β-cell recovery were associated with decreases liver and pancreas fat content following weight loss [32,33]. Chronic β-cell exposure to excess fat has been long observed to induce β-cell damage and loss of function before incident T2D. With hyperglycaemia becoming chronic, β-cell stress is further perpetuated [34]. Caloric restriction and weight loss have been associated with reduction of ectopic fat, including liver triglycerides content, which in turn has been associated with restoration of β-cell function and increased hepatic insulin sensitivity, possibly reverting T2D [33,35,36]. Accordingly, our results show that most subjects who reverted to normoglycemia had normal fat liver content, and, consistent with the FLI equation, which includes BMI, WC and plasma TG levels as variables, they also presented lower intrabdominal fat and a better metabolic profile than subjects who did not revert. Additionally, these subjects also performed PA ≥150 min/week and ate fruits and vegetables daily, which are in turn associated with lower body weight and BMI [37,38], and a low metabolic risk profile including NAFLD [35]. Importantly, a FLI score <30, increased physical activity, and healthy dietary habits were independent factors associated to prediabetes reversion, suggesting that absence of NAFLD and healthier lifestyle habits are protective of the progression to incident T2D.

Lastly, the ROC curve showed that FLI predicted prediabetes reversion with better accuracy than FPG. Taken together with the rest of our results and previous observations, FLI may represent an ideal predictor for prediabetes reversion as it relates to intraabdominal fat accumulation and the cardiometabolic state of the subject, which affect β-cell function and reduce insulin sensitivity time before the manifestation of hyperglycaemia, making FPG a less reliable predictor.

Additionally, authors would like to point out that the term "prediabetes" should not be used to describe a definitive condition for a series of reasons. Firstly, there is no global consensus on which criteria define the condition; moreover, given that a significant percentage of people with moderately high blood glucose will not progress to T2D and will instead experience a reversion to normoglycemia, it may result inappropriate to label the subject as "prediabetic". Prediabetes is not a diagnosis but rather a risk factor for T2D, and, most importantly, a reversible one.

Strengths and limitations of the study should be considered. The main strengths are the large sample size and the long follow-up period. Furthermore, the study population was representative of the Spanish workforce. Limitations, on the other hand, relate to possible misclassification bias as subjects were categorized as having prediabetes based on a single blood sample, thus limiting the possibility to account for intra-individual variability and increasing the possibility of a regression-toward-the-mean effect, possibly affecting the regression rate. Moreover, HbA1c values, as other criteria for the diagnosis of prediabetes, were not available for the whole sample. HbA1c has a lower variability that FPG [39], however, the study sample includes data from occupational routine health assessments, during which HbA1c measurements are done exclusively on workers with previously identified elevated FPG. Finally, lifestyle data (PA, diet, smoking habits, etc.) were not recorded at 5-year follow-up.

## Conclusions

Our current findings showed that a low FLI score is associated with prediabetes reversion at 5 years follow-up. Concretely, regular physical activity, healthy dietary habits and absence of hepatic steatosis are independently associated with the probability of reversion to

normoglycemia in adult workers with prediabetes at baseline. Since NAFLD is a risk factor for the development of T2D, the absence of hepatic steatosis may increase the chances of reverting from prediabetes to normoglycemia, especially in active subjects with healthy eating habits. Low FLI values (especially FLI< 30) may be useful to predict the probability of prediabetes reversion, especially in active subjects with healthy eating habits. Further studies could evaluate the role of key lifestyle habits in the reversion of prediabetes and adequately target those individuals who could benefit from an early intervention.

## Acknowledgments

The authors are grateful to the field staff and participants of the study.

## Author Contributions

**Conceptualization:** Carla Busquets-Cortés, Miquel Bennasar-Veny, Ángel Arturo López-González, Aina M. Yáñez.

**Data curation:** Carla Busquets-Cortés, Miquel Bennasar-Veny, Ángel Arturo López-González, Sergio Fresneda.

**Formal analysis:** Carla Busquets-Cortés, Miquel Bennasar-Veny, Aina M. Yáñez.

**Investigation:** Ángel Arturo López-González, Sergio Fresneda, Manuela Abbate.

**Methodology:** Miquel Bennasar-Veny, Aina M. Yáñez.

**Writing – original draft:** Carla Busquets-Cortés, Manuela Abbate.

**Writing – review & editing:** Miquel Bennasar-Veny, Ángel Arturo López-González, Sergio Fresneda, Aina M. Yáñez.

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
