## [Decision Letter · Decision Letter 0]

11 Feb 2021

PONE-D-20-38894

Utility of Fatty Liver Index to predict reversion to normoglycemia in people with prediabetes

PLOS ONE

Dear Dr. Bennasar-Veny,

Thank you for submitting your manuscript to PLOS ONE. After careful consideration, we feel that it has merit but does not fully meet PLOS ONE’s publication criteria as it currently stands. Therefore, we invite you to submit a revised version of the manuscript that addresses the points raised during the review process.

There is currently unprecedented pressure on reviewers. As such, I have been unable to obtain a timely second review of your manuscript. However, I have received a thoughtful, detailed review which can guide you in the revision of your manuscript. I agree with the reviewers comments.

We look forward to receiving your revised manuscript.

Kind regards,

Michael W Greene, Ph.D.

Academic Editor

PLOS ONE

Reviewers' comments:

Reviewer's Responses to Questions

**Comments to the Author**

1. Is the manuscript technically sound, and do the data support the conclusions?

Reviewer #1: Yes

2. Has the statistical analysis been performed appropriately and rigorously? 

Reviewer #1: Yes

3. Have the authors made all data underlying the findings in their manuscript fully available?

Reviewer #1: Yes

4. Is the manuscript presented in an intelligible fashion and written in standard English?

Reviewer #1: Yes

5. Review Comments to the Author

Reviewer #1: This is a well-designed observational prospective study about well-defined clinical question, the reversion of prediabetes according to baseline FLI levels.

The article is well structured and written, so I have no major objections, except that perhaps some comments should be made in the introduction, or even in the discussion, on the results of a previous study published in 2018 in PlosOne referring to a Spanish primary care population (The PREDAPS study): Franch-Nadal J, Caballeria L, Mata-Cases M, Mauricio D, Giraldez-García C, Mancera J, Goday A, Mundet-Tudurí X, Regidor E; PREDAPS Study Group. Fatty liver index isa predictor of incident diabetes in patients with prediabetes: The PREDAPS study. PLoS One. 2018 Jun 1;13(6):e0198327. doi: 10.1371/journal.pone.0198327. eCollection 2018.

In relation to the limitations of the study, it is worth mentioning the lack of HbA1c values, which is one of the diagnoses of prediabetes criteria (ADA) and due to its lower variability than Fasting Plasma Glucose is a more precise tool for diagnosis. Moreover, the fact that only one measurement of Fasting Plasma Glucose was available reduces accuracy of the diagnosis and precludes the exclusion of patients with an isolated elevated value that would disappear due to the regression-to-the-mean effect. I think that this limitation needs to be mentioned to a greater extent in the discussion.

Additionally, there are some minor corrections or suggestions that are detailed below.

Minor changes/corrections or suggestions

Abstract

1. Page 8 (lines 39): To homogenize with other OR values, add the third decimal: 1.544

Introduction:

2. page 10, line 86: Refer or add a short comment on the reference I have mentioned before (PREDAPS study).

3. Page 13, line 151. Correct NALF, it has to be NAFLD

Results:

4. Page 11, Table 1. There is an error on the WC line

WC (cm) 44.2 ± 53.7 88.9 ± 9.7 83.2 ± 9.2 < 0.001

Obviously, the mean of WC for all patients (44.2) must be between 83 and 89 cm. Note that the DE is also wrong and should be between 9.2 and 9.7

5. Page 16, line 220: To homogenize with other OR values, add the third decimal: 1.544

6. Page 18, lines 238-239 Remove the title of the figure as it is already at the end of de main text.

Discussion:

7. Page 20, line 301: I think that the limitations I suggested before (lack of HbA1c values and only one measurement of FPG) need to be commented to a greater extent here. An HbA1c between 5.7 and 6.4% is one of the criteria for the diagnoses of prediabetes (ADA) and due to its lower variability than Fasting Plasma Glucose is a more precise tool for diagnosis. Moreover, the fact that only one measurement of Fasting Plasma Glucose was available reduces accuracy of the diagnosis and precludes the exclusion of patients with an isolated elevated value that would disappear due to the regression-to-the-mean effect.

8. Page 13, line 20. Please, consider to add a comment or comparison with the results of the reference I have suggested before (the PREDAPS study)

Conclusions

9. The conclusions section does not literally correspond to the sentences included in the conclusions in the abstract. It must be homogenized, adding the phrases included in the abstract in this section.

10. Moreover, one of the sentences (Pag 20, line 307) is difficult to understand and has to be modified:

“Since NAFLD is a risk factor for the development of T2D, a reduced risk of presenting hepatic steatosis may increase the chances of reverting from prediabetes to normoglycemia…”

The authors cannot speculate about the reduction of the risk of hepatic steatosis on the reversion of prediabetes… So, I suggest to change the “a reduced risk of presenting” for “the absence of” or something like that.

References

Please add the PREDAPS study reference and complete or correct the following references:

11. Pag 21, line 332: remove prediabetes at the end of the ref

12. Pag 22, line 335: it has to be: BMJ. 2016 Nov 23;355:i5953. doi: 10.1136/bmj.i5953.

13. Pag 22, line 351: it has to be: Int. J. Environ. Res. Public Health 2019, 16, 3104; doi:10.3390/ijerph16173104

14. Pag 25, line 457: it has to be corrected and completed: Beta-cell function in obesity: effects of weight loss. Diabetes 2004 Dec;53 Suppl 3:S26-33. doi: 10.2337/diabetes.53.suppl_3.s26.

In summary, I want to congratulate the authors for the quality of the article. Finally, some comments in the discussion as I have suggested could be of great interest and well received.

6. PLOS authors have the option to publish the peer review history of their article (what does this mean?). If published, this will include your full peer review and any attached files.

Reviewer #1: No

---

## [Author Response · Author response to Decision Letter 0]

21 Feb 2021

Responses to reviewers:

Reviewer #1: This is a well-designed observational prospective study about well-defined clinical question, the reversion of prediabetes according to baseline FLI levels.

The article is well structured and written, so I have no major objections, except that perhaps some comments should be made in the introduction, or even in the discussion, on the results of a previous study published in 2018 in PlosOne referring to a Spanish primary care population (The PREDAPS study): Franch-Nadal J, Caballeria L, Mata-Cases M, Mauricio D, Giraldez-García C, Mancera J, Goday A, Mundet-Tudurí X, Regidor E; PREDAPS Study Group. Fatty liver index isa predictor of incident diabetes in patients with prediabetes: The PREDAPS study. PLoSOne. 2018 Jun 1;13(6):e0198327. doi: 10.1371/journal.pone.0198327. eCollection 2018.

In relation to the limitations of the study, it is worth mentioning the lack of HbA1c values, which is one of the diagnoses of prediabetes criteria (ADA) and due to its lower variability than Fasting Plasma Glucose is a more precise tool for diagnosis. Moreover, the fact that only one measurement of Fasting Plasma Glucose was available reduces accuracy of the diagnosis and precludes the exclusion of patients with an isolated elevated value that would disappear due to the regression-to-the-mean effect. I think that this limitation needs to be mentioned to a greater extent in the discussion.

Additionally, there are some minor corrections or suggestions that are detailed below.

Minor changes/corrections or suggestions

Abstract

1. Page 8 (lines 39): To homogenize with other OR values, add the third decimal: 1.544. 

R: We thank the reviewer for pointing out the mistake. We have now replaced 1.54 with 1.544 in the abstract. 

Introduction:

2. page 10, line 86: Refer or add a short comment on the reference I have mentioned before (PREDAPS study). 

R: We thank the referee for the useful suggestion. We have commented in the introduction results from the article mentioned.

3. Page 13, line 151. Correct NALF, it has to be NAFLD. 

R: We have corrected the typo.

Results:

4. Page 11, Table 1. There is an error on the WC line WC (cm) 44.2 ± 53.7 88.9 ± 9.7 83.2 ± 9.2 < 0.001. Obviously, the mean of WC for all patients (44.2) must be between 83 and 89 cm. Note that the DE is also wrong and should be between 9.2 and 9.7. 

R: We are grateful to the reviewer for pointing out this inaccuracy, and we have replaced the mistaken information with correct data.

5. Page 16, line 220: To homogenize with other OR values, add the third decimal: 1.544. 

R: We have replaced 1.54 with 1.544 in the results, according to the reviewer’s suggestion.

6. Page 18, lines 238-239 Remove the title of the figure as it is already at the end of de main text. R: We removed the title of the figure within the text, as it is at the end of the main text.

Discussion:

7. Page 20, line 301: I think that the limitations I suggested before (lack of HbA1c values and only one measurement of FPG) need to be commented to a greater extent here. An HbA1c between 5.7 and 6.4% is one of the criteria for the diagnoses of prediabetes (ADA) and due to its lower variability than Fasting Plasma Glucose is a more precise tool for diagnosis. Moreover, the fact that only one measurement of Fasting Plasma Glucose was available reduces accuracy of the diagnosis and precludes the exclusion of patients with an isolated elevated value that would disappear due to the regression-to-the-mean effect. 

R: We thank the reviewer for the comment. We agree that HbA1c has a lower variability that Fasting Plasma Glucose (FPG), however, as now explained as a limitation, this variable was not collected for all subjects. The study sample includes data from occupational routine health assessments, during which HbA1c measurements are done on workers with previously identified elevated FPG. 

We also agree with the fact that with only one measurement of FPG available, the possibility to regression toward the mean phenomenon can take place, thus possibly affecting the regression rate. We included such limitation in the text. 

8. Page 13, line 20. Please, consider to add a comment or comparison with the results of the reference I have suggested before (the PREDAPS study). 

R: We thank the referee for the useful suggestion. We have commented in the discussion results from the article mentioned. 

Conclusions:

9. The conclusions section does not literally correspond to the sentences included in the conclusions in the abstract. It must be homogenized, adding the phrases included in the abstract in this section. 

R: We thank the reviewer for the valuable comment. Accordingly, we have modified the conclusion within the text in order to homogenize it with the conclusions in the abstract.

10. Moreover, one of the sentences (Pag 20, line 307) is difficult to understand and has to be modified:

“Since NAFLD is a risk factor for the development of T2D, a reduced risk of presenting hepatic steatosis may increase the chances of reverting from prediabetes to normoglycemia…”

The authors cannot speculate about the reduction of the risk of hepatic steatosis on the reversion of prediabetes… So, I suggest to change the “a reduced risk of presenting” for “the absence of” or something like that. 

R: We agree with the reviewer comment. We removed “a reduced risk of presenting” and wrote “the absence of” instead.

References

Please add the PREDAPS study reference and complete or correct the following references:

11. Pag 21, line 332: remove prediabetes at the end of the ref. 

R: Thank you for pointing out the mistake. We removed the word “prediabetes” at the end of the reference.

12. Pag 22, line 335: it has to be: BMJ. 2016 Nov 23;355: i5953. doi: 10.1136/bmj.i5953. 

R: We have corrected the reference.

13. Pag 22, line 351: it has to be: Int. J. Environ. Res. Public Health 2019, 16, 3104; doi:10.3390/ijerph16173104. 

R: We have added the missing information.

14. Pag 25, line 457: it has to be corrected and completed: Beta-cell function in obesity: effects of weight loss. Diabetes 2004 Dec;53 Suppl 3:S26-33. doi: 10.2337/diabetes.53.suppl_3.s26. 

R: We have completed the information. The title was missing due to an error with our cite-o-matic system.

In summary, I want to congratulate the authors for the quality of the article. Finally, some comments in the discussion as I have suggested could be of great interest and well received.

R: We thank the reviewer for the kind comments and for giving us the opportunity to revisit the manuscript while taking into consideration the comments made.

---

## [Decision Letter · Decision Letter 1]

15 Mar 2021

Utility of Fatty Liver Index to predict reversion to normoglycemia in people with prediabetes

PONE-D-20-38894R1

Dear Dr. Bennasar-Veny,

We’re pleased to inform you that your manuscript has been judged scientifically suitable for publication and will be formally accepted for publication once it meets all outstanding technical requirements.

Kind regards,

Michael W Greene, Ph.D.

Academic Editor

PLOS ONE

Reviewers' comments:

Reviewer's Responses to Questions

**Comments to the Author**

1. If the authors have adequately addressed your comments raised in a previous round of review and you feel that this manuscript is now acceptable for publication, you may indicate that here to bypass the “Comments to the Author” section, enter your conflict of interest statement in the “Confidential to Editor” section, and submit your "Accept" recommendation.

Reviewer #1: All comments have been addressed

2. Is the manuscript technically sound, and do the data support the conclusions?

Reviewer #1: Yes

3. Has the statistical analysis been performed appropriately and rigorously? 

Reviewer #1: Yes

4. Have the authors made all data underlying the findings in their manuscript fully available?

Reviewer #1: Yes

5. Is the manuscript presented in an intelligible fashion and written in standard English?

Reviewer #1: Yes

6. Review Comments to the Author

Reviewer #1: (No Response)

7. PLOS authors have the option to publish the peer review history of their article (what does this mean?). If published, this will include your full peer review and any attached files.

Reviewer #1: No

---

## [Editor Report · Acceptance letter]

19 Mar 2021

PONE-D-20-38894R1 

Utility of Fatty Liver Index to predict reversion to normoglycemia in people with prediabetes. 

Dear Dr. Bennasar-Veny:

I'm pleased to inform you that your manuscript has been deemed suitable for publication in PLOS ONE. Congratulations! Your manuscript is now with our production department. 

Kind regards, 

on behalf of

Dr. Michael W Greene 

Academic Editor

PLOS ONE